# Familiarity-Based Open-Set Recognition Under Adversarial Attacks

Philip Enevoldsen,* Christian Gundersen,* Nico Lang, Serge Belongie, and Christian Igel

Department of Computer Science, University of Copenhagen

## Abstract

Open-set recognition (OSR), the identification of novel categories, can be a critical component when deploying classification models in real-world applications. Recent work has shown that familiarity-based scoring rules such as the Maximum Softmax Probability (MSP) or the Maximum Logit Score (MLS) are strong baselines when the closed-set accuracy is high. However, one of the potential weaknesses of familiarity-based OSR are adversarial attacks. Here, we study gradient-based adversarial attacks on familiarity scores for both types of attacks, False Familiarity and False Novelty attacks, and evaluate their effectiveness in informed and uninformed settings on TinyImageNet. Furthermore, we explore how novel and familiar samples react to adversarial attacks and formulate the adversarial reaction score as an alternative OSR scoring rule, which shows a high correlation with the MLS familiarity score.

## 1 Introduction

In many real-world applications of machine learning models, it is crucial to understand the models' limitations and the trustworthiness of their predictions in novel situations. Thus, we investigate open-set recognition (OSR) [1], which can be seen as a special case of out-of-distribution (OOD) detection [2]. The OSR task is to identify novel categories, which were not included in the training dataset, at test time. Recently, Vaze et al. [3] have demonstrated that the progress in OSR performance over the past years is not necessarily due to advancement in OSR approaches, but is correlated with improved performance on the closed-set categories, i.e., the classification of categories included in the training dataset. With this observation, simple baseline scoring rules such as the Maximum Softmax Probability (MSP) [4] and the Maximum Logit Score (MLS) [3, 5] are competitive and perform on par with—or even outperform—more dedicated approaches such as ARPL and ARPL+CS [6], OSRCI [7], and OpenHybrid [8]. At the same time, Dieterich & Guyer [9] have proposed the Familiarity Hypothesis, stating that such familiarity-based scoring rules identify novel categories by measuring the absence

---

*Equal contribution.

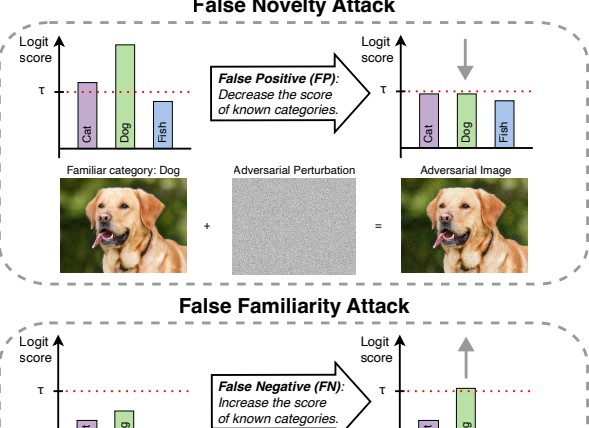

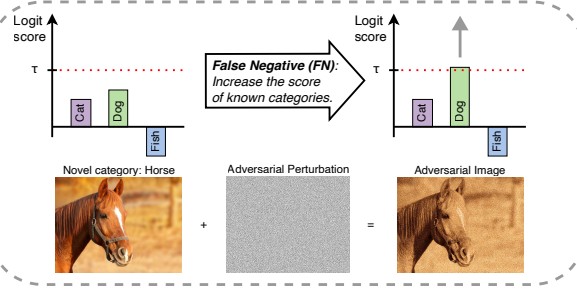

**Figure 1. Adversarial attacks on OSR familiarity scores.** Considering novel categories as positives, the top box depicts a *false positive* (FP) attack that lowers the familiarity of the known category leading to a *false novelty* (FNov). In contrast, the bottom box indicates a *false negative* (FN) attack that increases the familiarity of a known category leading to a *missed novelty* or *false familiarity* (FFam).

of familiar features instead of actively recognizing the presence of novel features. They investigated occlusions as one of the weaknesses of familiarity-based OSR, which can cause false novelty detections. Adversarial attacks pose another potential weakness to familiarity-based OSR, which we study in this work. Dietterich & Guyer [9] mention the risks of adversarial vulnerability in their outlook discussion:

> "By applying existing attack algorithms (e.g., the FGSM [10]), we predict that it will be very easy to raise the logit score of at least one class and thereby hide a novel class image from novelty detection. It may also be possible to depress the logit scores of enough classes to create false anomaly alarms."

In other words, this prediction states that it might be easy to compute adversarial perturbations that amplify familiar features to cause a *false familiarity*

Proceedings of the 6th Northern Lights Deep Learning Conference (NLDL), PMLR 265, 2025.

(FFam), but it might be harder to hide (all) familiar features to yield a *false novelty* (FNov) (see Fig. 1). While it has been shown that the OSR approach OpenMax [11] is vulnerable to adversarial attacks [12, 13], we study the vulnerability of familiarity-based OSR approaches to gradient-based adversarial white-box attacks (i.e., the model parameters are given).

A crucial difference to prior work studying adversarial robustness of OOD detection is that familiarity-based OSR approaches neither train on auxiliary OOD data (like, e.g., [14, 15]), nor introduce an explicit category for the "openness" of the test data (e.g., [11]). Consequently, this requires different designs of adversarial objectives. Ultimately, using such perturbations for adversarial training may yield better robustness, but prior work relies on training with both perturbed closed-set and auxiliary OOD data [16].

Although we do not use the presented objectives for adversarial training in this work, we explore whether these adversarial attacks can be used to design an alternative OSR score. Liang et al. [17] found that OOD detection was enhanced by pre-processing inputs with an adversarial perturbation. We explore if this generalizes to the OSR setting and whether it can be used to develop new OSR methods. In summary, this study aims to answer four main questions:

1. **False Familiarity vs. False Novelty:** What type of attack is more effective?

2. **Uninformed vs. informed attacks:** How can adversarial attacks profit from knowing the input type (i.e., closed-set or open-set sample)?

3. **FGSM vs. iterative attacks:** Can more flexible iterative attacks improve upon the fast gradient sign (FGSM) method?

4. **OSR scores using adversarial attacks:** Can we use the reaction to adversarial perturbations as an OSR score to separate familiar and novel samples?

# 2 Methodology

## 2.1 Familiarity-based open-set recognition (OSR)

We consider an input space $\mathcal{X}$ and a set $\mathcal{F}$ of *familiar* categories, i.e., the closed-set. In closed-set recognition (CSR), the objective is to model the probability $p(y \mid \boldsymbol{x}, y \in \mathcal{F})$, where $y$ is a label that is associated with the input $\boldsymbol{x} \in \mathcal{X}$. The model is trained on a training dataset $\mathcal{D}_{\text{train}} = \{(\boldsymbol{x}_i, \boldsymbol{y}_i)\}_{i=1}^N \subset \mathcal{X} \times \mathcal{F}$ and evaluated on a non-overlapping closed-set test set, $\mathcal{D}_{\text{test-csr}} = \{(\boldsymbol{x}_i, \boldsymbol{y}_i)\}_{i=1}^M \subset \mathcal{X} \times \mathcal{F}$ that contains the

categories given at train time. We consider a deep neural network $f_{\boldsymbol{\theta}} : \mathcal{X} \to \mathbb{R}^{|\mathcal{F}|}$ parameterized by $\boldsymbol{\theta}$ for modelling $p(y \mid \boldsymbol{x}, y \in \mathcal{F})$. Here, $f_{\boldsymbol{\theta}}$ maps an input to a vector of logits that are normalized using the softmax function $\sigma : \mathbb{R}^{|\mathcal{F}|} \to (0, 1)^{|\mathcal{F}|}$ to obtain pseudo-probabilities for the familiar categories.

In open-set recognition (OSR) a set $\mathcal{N}$ of *novel* categories is additionally considered and a test set containing inputs from both novel and familiar classes is used to evaluate the OSR performance: $\mathcal{D}_{\text{test-osr}} = \{(\boldsymbol{x}_i, \boldsymbol{y}_i)\}_{i=1}^M \subset \mathcal{X} \times (\mathcal{F} \cup \mathcal{N})$. A balanced test set containing an equal number of familiar and novel samples is typically used to evaluate the OSR performance. To decide whether $y \in \mathcal{F}$, a familiarity score, $\mathcal{S}(y \in \mathcal{F} \mid \boldsymbol{x})$, is calculated and used to rank the test samples in $\mathcal{D}_{\text{test-osr}}$. Familiarity-based scoring rules include the *Maximum Softmax Probability (MSP)* score [4]

$$\mathcal{S}_{\text{MSP}}(y \in \mathcal{F} \mid \boldsymbol{x}) = \max_y \sigma(f_{\boldsymbol{\theta}}(\boldsymbol{x}))_y \qquad (1)$$

and the *Maximum Logit Score (MLS)* [3, 9]

$$\mathcal{S}_{\text{MLS}}(y \in \mathcal{F} \mid \boldsymbol{x}) = \max_y f_{\boldsymbol{\theta}}(\boldsymbol{x})_y, \qquad (2)$$

which has outperformed the MSP score in prior work [3]. For both scoring rules, high scores indicate familiar and low scores indicate novel categories.

## 2.2 Fast gradient sign method

A simple and effective method for generating adversarial inputs is the *Fast Gradient Sign Method* (FGSM) which was first described in [10]. The FGSM generates an adversarial input, $\boldsymbol{x}^{\text{adv}}$, using the following rule:

$$\boldsymbol{x}^{\text{adv}} = \boldsymbol{x} + \varepsilon \, \text{sign}[\nabla_{\boldsymbol{x}} \mathcal{L}(\boldsymbol{\theta}, \boldsymbol{x}, y)] \qquad (3)$$

Here $\boldsymbol{x}$ represents the unmodified input and the second term is known as the *adversarial perturbation*, where $\varepsilon$ controls the magnitude of the perturbation (as visualized in Fig. 2). Initially, $\mathcal{L}$ is set to the training objective [10], but can be any objective function that an adversary aims to optimize. The adversarial perturbation is constrained such that $\|\boldsymbol{x}^{\text{adv}} - \boldsymbol{x}\|_\infty \leq \varepsilon$.

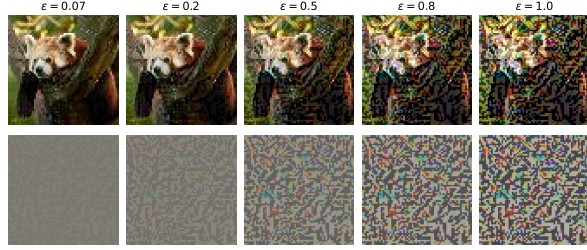

**Figure 2. Qualitative example.** Perturbed images (top) and adversarial perturbations (bottom).

## 2.3 Iterative attacks

Iterative approaches can generate more diverse perturbations compared to the FGSM by optimizing the objective function in a more flexible manner but at higher computational costs. An iterative adversarial attack takes the general form

$$\boldsymbol{x}_{n+1}^{\mathrm{adv}} = \mathrm{Clip}_{\boldsymbol{x},\varepsilon}\{\boldsymbol{x}_n^{\mathrm{adv}} + \mathrm{Step}(\mathcal{L},\boldsymbol{\theta},\boldsymbol{x}_n^{\mathrm{adv}},y)\} \ , \quad (4)$$

starting from $\boldsymbol{x}_0^{\mathrm{adv}} = \boldsymbol{x}$, where $\mathrm{Clip}_{\boldsymbol{x},\varepsilon}(\boldsymbol{z})_i = \min(\max(z_i, x_i - \varepsilon), x_i + \varepsilon)$. The *Basic Iterative Method (BIM)* [18] applies the FGSM update iteratively by setting $\mathrm{Step}(\mathcal{L},\boldsymbol{\theta},\boldsymbol{x}_n^{\mathrm{adv}},y) = \alpha \ \mathrm{sign}(\nabla_{\boldsymbol{x}}\mathcal{L}(\boldsymbol{\theta},\boldsymbol{x}_n^{\mathrm{adv}},y))$. In this method, the step size $\alpha$ and the number of iterations can be adjusted to get the desired trade-off between run-time and performance. Alternative approaches are inspired by gradient-based optimizers using momentum to improve performance [19]. We investigate an iterative approach using RPROP [20–22] that relaxes the fixed step size $\alpha$ of BIM with an adaptive step size, where $\mathrm{Step}(\mathcal{L},\boldsymbol{\theta},\boldsymbol{x}_n^{\mathrm{adv}},y)$ denotes the update step computed by some iterative optimization method. RPROP adjusts the step size separately for each optimizable parameter while iterating—in the case of adversarial attacks on images for every pixel per channel. Hence, adversarial perturbations created with RPROP can be sparse and may therefore be less noticeable. For a fair comparison with FGSM, the perturbations are constraint to $\|\boldsymbol{x}^{\mathrm{adv}} - \boldsymbol{x}\|_\infty \le \varepsilon$.

## 2.4 Adversarial attacks on familiarity-based OSR

The goal of adversarial attacks on OSR is to destroy the ranking of novel and familiar samples given by the OSR score, e.g., MLS or MSP. We discuss two types of attacks as illustrated in Fig. 1.

**False Familiarity (FFam).** False Familiarity attacks aim to *increase* the logit (or softmax probability) of an arbitrary familiar category, which is similar to targeted attacks in closed-set recognition [23]. We investigate three objective functions to achieve this attack:

$$\mathcal{L}_{\max}(\boldsymbol{\theta},\boldsymbol{x},y) = \max_{y'} f_{\boldsymbol{\theta}}(\boldsymbol{x})_{y'} \quad (5)$$

$$\mathcal{L}_{2\text{-norm}}(\boldsymbol{\theta},\boldsymbol{x},y) = \|f_{\boldsymbol{\theta}}(\boldsymbol{x})\|_2 \quad (6)$$

$$\mathcal{L}_{\text{log-MSP}}(\boldsymbol{\theta},\boldsymbol{x},y) = \log \max_{y'} \sigma(f_{\boldsymbol{\theta}}(\boldsymbol{x}))_{y'} \quad (7)$$

The log-MSP loss has been proposed in the ODIN approach [17] (which was refined in the generalized ODIN [24]) to preprocess images with adversarial perturbations to improve OOD detection using the MSP score.[1]

---

[1]While this is not further investigated in this work, our OSR experiments did not confirm an improvement over the MSP score as also mentioned in other independent work [9].

**False Novelty (FNov).** In this likely more challenging setting, we may have to decrease the logits of multiple categories either with a single FGSM step or multiple iterative steps. Objective functions rewarding only the decrease of the largest logit might fail, thus, besides the $\mathcal{L}_{\max}$, we investigate a the $\mathcal{L}_{2\text{-norm}}$ and the sum-exp loss:

$$\mathcal{L}_{\text{sum-exp}}(\boldsymbol{\theta},\boldsymbol{x},y) = \sum_{y' \in |\mathcal{F}|} e^{f_{\boldsymbol{\theta}}(\boldsymbol{x})_{y'}} \quad (8)$$

The 2-norm encourages reducing non-maximum logits while still prioritizing the max logit. However, one limitation of the 2-norm is that it is non-negative. Since logits are unnormalized and can be negative, it would be preferable if the objective function also rewarded making the logits negative. This led us to propose the sum-exp loss, which continues to decrease if a logit becomes negative. Importantly, while False Familiarity attacks *maximize* these objectives, False Novelty attacks *minimize* them.

## 2.5 Uninformed vs. informed attacks

We call an attack *informed* if the adversary has access to the binary set-labels of the input, i.e., closed-set vs. open-set, and *uninformed* if that information is not available [25]. In the uninformed setting, either a FNov or FFam attack is applied to all images, disregarding whether an image is novel or familiar. For informed adversaries, FFam attacks are only performed on novel images to make the classifier falsely predict that the inputs are familiar, leaving all familiar images unchanged. In contrast, FNov attacks are only performed on familiar images to make the classifier wrongly assume novel inputs, leaving all novel images unchanged.

## 2.6 Adversarial reaction score (ARS)

The idea that novel and familiar inputs could react differently to adversarial attacks is motivated by the findings in [17] as discussed in the introduction and by our own analyses of the behaviour of scores before and after adversarial perturbation (see Fig. 5). Here, we present the concept of an Adversarial Reaction Score (ARS) as an alternative OSR score. The ARS measures the reaction to an adversarial attack applied to a given input. We define the ARS as the signed difference between the MLS (or MSP) before and after the adversarial attack:

$$\mathcal{S}_{\mathrm{adv}}(y \in \mathcal{F} \mid \boldsymbol{x}) = \max_y f_{\boldsymbol{\theta}}(\boldsymbol{x}^{\mathrm{adv}})_y - \max_y f_{\boldsymbol{\theta}}(\boldsymbol{x})_y \quad (9)$$

Note that this may involve using logits (or probabilities) from different categories to calculate a score.

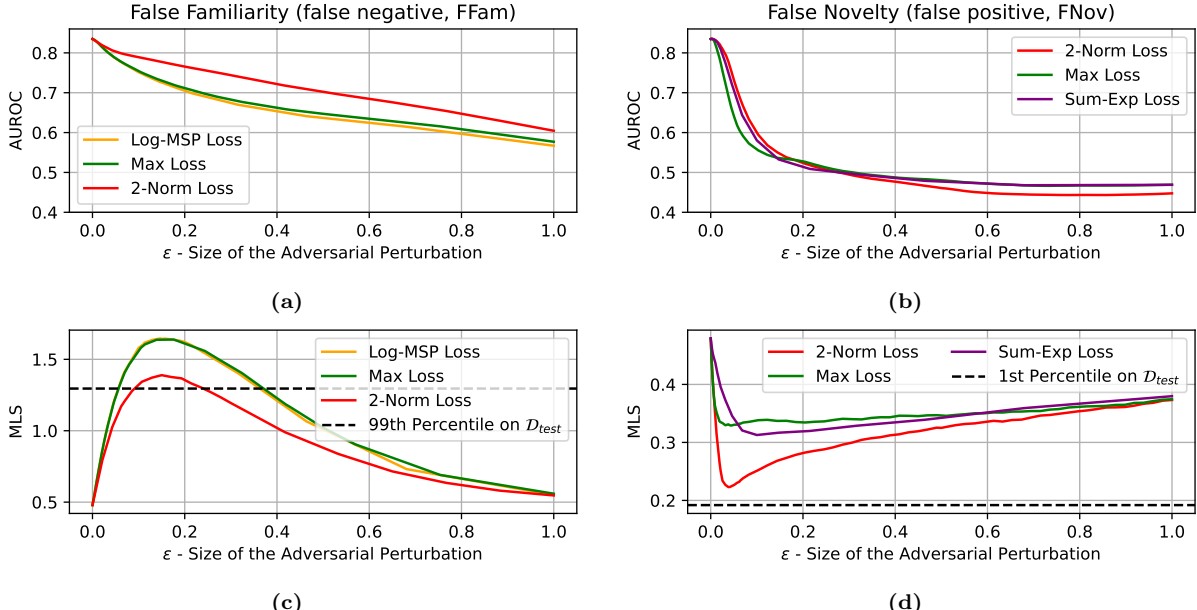

**Figure 3. Uninformed FGSM attacks.** Fast gradient sign method (FGSM) attacks on TinyImageNet. Left: False Familiarity (FFam) attacks. Right: False Novelty (FNov) attacks. (a,b) The OSR ranking measured by AUROC. (c,d) Median Maximum Logit Score (MLS) of all samples (familiar and novel).

# 3 Experimental results

We experiment with the TinyImageNet dataset [26], described as one of the more challenging benchmarks used in the OSR literature [3]. Here we use the open-set split presented in Vaze et al. [3] and follow their experimental setup. TinyImageNet consists of a subset of 200 ImageNet categories [27], whereas 20 classes are used as the closed-set training dataset and 180 classes as the open-set. The CNN architecture was a VGG32[2], a lightweight version of the VGG architecture [28]. This results in a closed-set accuracy of 84.2% averaged over five class splits.

We report the OSR performance of the MLS for the first of the five splits with the area under the Receiver-Operator curve (AUROC). The AUROC is a threshold-less metric that evaluates the ranking from open-set to closed-set samples. As a higher AUROC means better OSR performance, adversarial attacks aim to lower the AUROC.

**What type of attack is more effective?** It depends. In the *uninformed* FGSM experiments, False Novelty attacks are more effective in destroying the ranking, i.e., decreasing the AUROC, than False Familiarity attacks at the same magnitude $\varepsilon$ of adversarial perturbation (Fig. 3(a), 3(b)). However, we observe the opposite in the *informed* FGSM setting (Fig. 4(a), 4(b)), which also holds for the informed iterative attack (Fig. 6), where the AUROC of FFam attacks is lower than FNov attacks. To understand

this behaviour, we look at the distribution of scores before and after the attacks (see Fig. 5).

**It is too easy to raise the logit score.** Or in other words, it is easy to amplify familiar features. While FFam attacks aim to amplify familiar features of the open-set to cause a missed novelty, FNov attacks aim to hide familiar features to reduce the familiarity of closed-set categories. We recall that uninformed attacks are performed on both novel and familiar images. Even though FFam attacks can increase the median MLS above the 99th percentile of the original test data scores (Fig. 3(c)), the AUROC is rather preserved (Fig. 3(a)). Hence, the FFam attacks not only increase the familiarity (i.e., MLS) of the novel but also of the familiar samples, which preserves the ranking (see also Fig. 5(a)). In contrast, FNov attacks cannot decrease the median MLS below the 1st percentile of the original test scores (Fig. 3(d)), but the ranking (AUROC) is effectively destroyed (Fig. 3(b)). This suggests that FNov attacks tend to decrease the scores of the closed-set more than the scores of the open-set (see Fig. 5(b)). Our experiments confirm the prediction of Dietterich & Guyer [9] that it is very easy to raise the logit score, which only leads to effective FFam attacks in the informed setting. However, our results reveal that for uninformed attacks the ability to easily raise the logit score is not the key to attack the ranking of familiarity-based OSR approaches.

**Which objective function performs best?** While some objectives are able to perform both types of attacks by swapping the sign, no objective

---

[2]We use the model weights published on: `https://github.com/sgvaze/osr_closed_set_all_you_need` (accessed 2023-05-23).

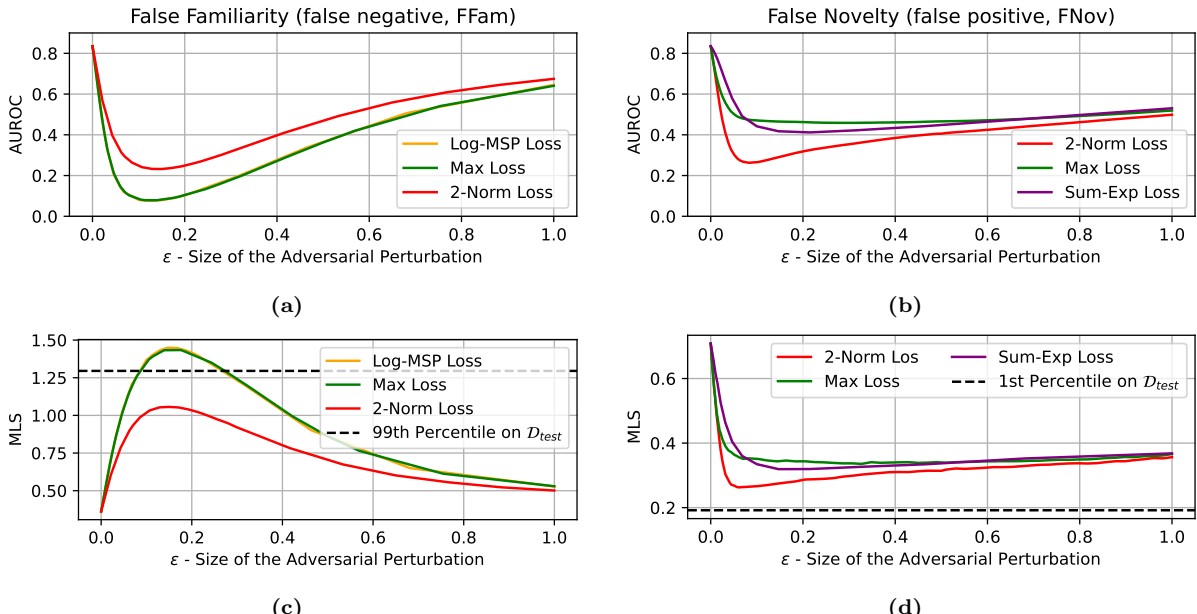

**(a)**       **(b)**

**(c)**       **(d)**

**Figure 4. Informed FGSM attacks.** Fast gradient sign method (FGSM) attacks on TinyImageNet. Left: False Familiarity (FFam) attacks. Right: False Novelty (FNov) attacks. (a,b) The OSR ranking measured by AUROC. (c,d) Median Maximum Logit Score (MLS) of novel samples (c) and familiar samples (d).

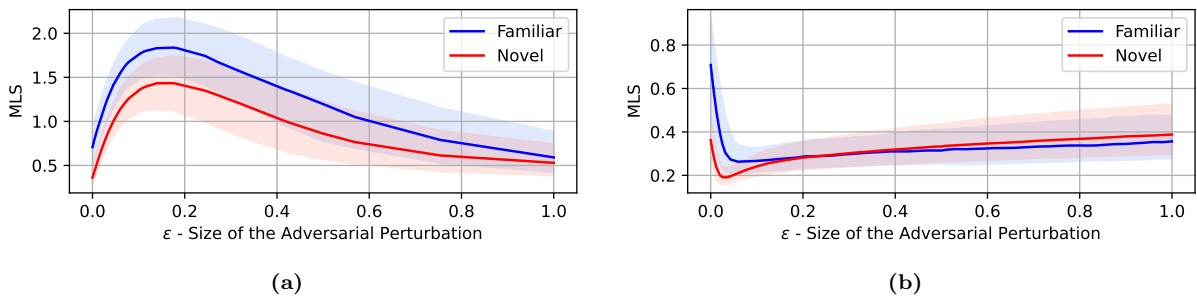

**(a)**       **(b)**

**Figure 5. MLS for familiar and novel samples after adversarial perturbation.** FGSM attacks on TinyImageNet. Median Maximum Logit Score (MLS) w.r.t. original scores separately for the familiar and novel samples. The filled region shows the 25th and 75th quantile. (a) False Familiarity (FFam) attacks with the max loss. (b) False Novelty (FNov) attacks with the 2-norm loss. These were the objectives that could push the scores most up or down w.r.t. the original scores, for FFam and FNov, respectively (see Fig. 3(c), 3(d)).

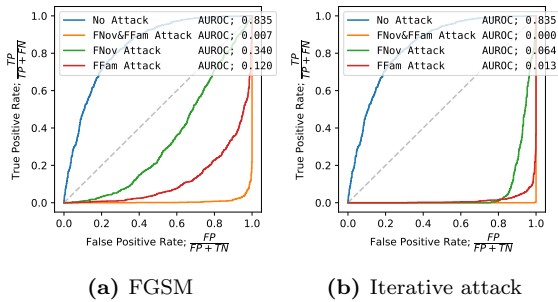

**(a)** FGSM       **(b)** Iterative attack

**Figure 6. Informed FGSM vs. iterative attacks.** False Novelty (FNov) attacks are performed on familiar samples (2-norm loss) and False Familiarity (FFam) attacks on novel samples (max loss). (a) Fast gradient sign method (FGSM) using $\varepsilon = 0.07$ for FFam and $\varepsilon = 0.04$ for FNov. (b) Our iterative method with $\varepsilon = 0.07$ for both the FFam and FNov attacks.

is clearly best for both FFam and FNov attacks (Fig. 3, 4). In the uninformed setting (Fig. 3), FFam attacks achieve the lowest AUROC using the Log-MSP loss and second lowest with the Max loss. For FNov attacks, the Max loss achieves the lowest AU-ROC with $\epsilon < 0.1$. Whereas at $\epsilon \approx 0.3$ all objective functions achieve an AUROC of $\approx 0.5$, for $\epsilon > 0.3$ the 2-norm achieves even lower AUROC.

**Informed attacks reverse the ranking almost perfectly.** As expected, informed FGSM attacks (Fig. 4) can improve substantially over uninformed attacks (Fig. 3). While the FFam attacks can increase the median MLS of the adversarial novel samples beyond the 99th percentile of the test dataset (Fig. 4(c)), the FNov attacks cannot decrease the median MLS of the adversarial familiar samples below the 1st percentile (Fig. 4(d)). To study the

combination of informed attack types in Fig. 6, we use the objectives yielding the lowest AUROC, i.e., the 2-norm for FNov and the max loss for FFam attacks. Ultimately, when using both FNov attacks on familiar and FFam attacks on novel samples together, informed attacks (both FGSM and iterative) are able to reverse the ranking of novel and familiar images almost perfectly (Fig. 6).

**FGSM vs. iterative attacks.** Informed iterative attacks are able to decrease the AUROC by an order of magnitude compared to informed FGSM attacks using the same or even smaller $\varepsilon$ (Fig. 6). The AUROC for FNov attacks is decreased from 0.34 (FGSM) to 0.06 (iterative) and for FFam attack from 0.12 (FGSM) to 0.01 (iterative).

**Can adversarial reaction scores be used for OSR?** We calculated the ARS as in Eq. 9 by obtaining $\boldsymbol{x}^{\mathrm{adv}}$ with the FGSM using all combinations of loss-functions (5)-(8), attack types (i.e., FFam and FNov) and $\varepsilon$-values. While all combinations exhibit similar behaviour, we focus only on the combination that achieved the best result. We found this to be the MLS based ARS using FNov FGSM with the 2-norm loss and $\varepsilon = 0.051$. This resulted in an AUROC of 0.81 compared to the 0.83 obtained by the MLS scoring rule. Statistics from the evaluation of our best ARS are shown in Fig. 7. Figure 7(a) shows that the ARS indeed allows to distinguish novel and familiar images. However, Fig. 7(b) reveals that the ARS is strongly correlated with the MLS of the original unattacked images. Given the same MLS value, the overlapping trend lines show that the ARS for a fixed MLS is very similar for novel and familiar images. Hence, the effectiveness of the ARS as an OSR score could be explained by the effectiveness of the MLS, but further analyses are needed to gain a better understanding of how the MLS and ARS are related.

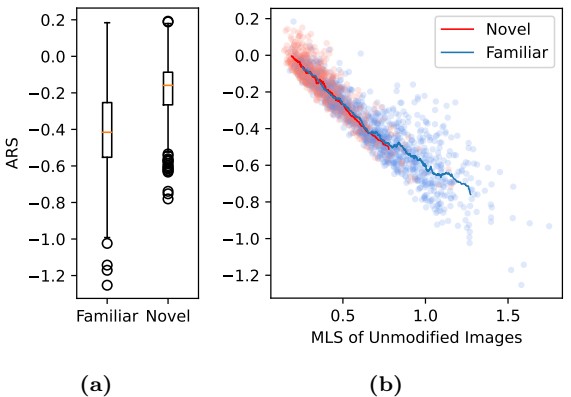

(a)                         (b)

**Figure 7. Adversarial reaction score (ARS).** This ARS uses the false novelty FGSM with the 2-norm loss and $\varepsilon = 0.051$. (a) ARS of familiar and novel samples. (b) ARS compared with MLS. The trend lines show a sliding average of the ARS of samples with similar MLS.

# 4    Limitations

While our study presents adversarial attacks on familiarity-based OSR scores and steps towards understanding their adversarial robustness, some limitations exist. Our empirical evaluation is limited to the TinyImageNet dataset and the VGG32 architecture. Further research is needed to determine if our findings generalize to other datasets and model architectures. Furthermore, we focused on the MLS and it needs to be tested if the observations hold for alternative familiarity scores, like the MSP.

While the studied objective functions are applicable to both MLS and MSP, the 2-norm can be affected by negative scores. For the FFam attack, where we aim to increase the max logit, the attack would fail if the logit with the largest absolute value is negative. In this case, the 2-norm would encourage making the negative logit even more negative. Empirically, this rarely occurred in our experiments. Finally, with very large epsilon values the adversarial perturbations are clearly visible in the input images (see Fig. 2). These noisy images lead to random OSR scores, which cause a random ranking of the test samples as measured by the AUROC. However, large epsilons do not lead to meaningful subtle adversarial attacks.

# 5    Conclusion

We have studied the vulnerability of familiarity-based OSR approaches to adversarial attacks. Our MLS experiments confirm Dietterich & Guyer's [9] prediction that the logit score can be easily raised with an adversarial perturbation. However, their prediction did not specify what information is available to the attack. Here, we show that this ability – to easily raise the logit score of one class – *only* leads to effective false familiarity attacks in the informed setting. In the uninformed setting, such attacks are less effective than false novelty attacks, which, in contrast, are able to successfully destroy the ranking by lowering the logit scores of closed-set categories. Furthermore, we explored using adversarial attacks for defining new OSR scores. However, our ARS scoring rules, contrary to the findings in [17], did not seem to improve upon the MLS. Nonetheless, we believe that our study can contribute to the design of better scoring rules in the future and to making familiarity scores robust against adversarial attacks.

**Acknowledgement** P.E. and C.G. acknowledge support by the Danish Data Science Academy (DDSA). This work was supported in part by the Pioneer Centre for AI, DNRF grant number P1, and the European Union project ELIAS (grant agreement number 101120237).

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
