# OpenReview forum: "Familiarity-Based Open-Set Recognition Under Adversarial Attacks"
_NLDL.org/2025/Conference — NLDL 2025 Oral_

### Official Review · Reviewer_6NYm · 2024-09-18
**Overall the submitted article is well-written and well explained**

**Confidence:** 4

**Summary:**

The paper mainly focuses on exploring how to evaluate the effect of adversarial attacks on the familiarity-based OSR. It discusses the effects of different metrics based on the type of attack.

The article is well-written and elaborates on the proposed approach to adversarial attacks. It discusses the impact on performance based on the metric selected for optimization.

The authors also proposed a new scoring method but it doesn't seem to improve upon the existing methods.

**Strengths:**

The authors have done a good job of elaborating and explaining different types of attacks as well as the different objective functions being used. Along with this, they also have done a good job of discussing the methodology in detail.

**Weaknesses:**

From the reviewer's perspective it feels more details could be added for the experiments and the explanation of the figures as it would improve the readability of the paper.

It is critical to include the future directions for the improvements and how the authors see their work contributing to the research in the longer term since the metrics proposed by the authors don't necessarily improve the existing ones.

**Justification:**

Overall, it is a well-written paper that elaborates on the necessary technical details of the proposed approach. The paper could be improved by adding more details to the experimental results section and highlighting the future directions for the work.

---

> ### Author Rebuttal · Authors · 2024-10-25
>
> Thank you for the feedback. We will aim to provide more details for the experiments where needed. It would be appreciated if the reviewer could point us to specific details that were not clear yet. Thanks!

---

### Official Review · Reviewer_tBQx · 2024-09-20
**Well motivated research, but there are concerns on the main claims**

**Confidence:** 4

**Summary:**

Open-set recognition (OSR) task is a special type of out-of-distribution (OOD) task. In OSR, we need to distinguish samples from novel classes (unseen during training) and familiar (known) classes. Recent works have demonstrated that familiarity-based scores can achieve competitive performances in OSR tasks. Familiarity-based scores use the maximum estimated probability of (known) classes (Maximum SoftMax Probability, MSP) or the maximum logit output of (known) classes (Maximum Logit Score, MLS). The hypothesis of Dietterich & Guyer can be summarized as "the logit score on only one class can be easily increased with an adversarial perturbation while it might be more difficult to decrease the logit score for all (known) classes." From this, the paper first speculates that it will be easier to cause false familiarity than to cause false novelty.

This research investigated adversarial attacks on MLS. In uninformed attacks, where the attacker does not know if a sample is known or novel, false novelty (and false familiarity) attacks are applied on both known and novel samples to destroy the familiarity ranking of samples. In informed attacks, where the attacker knows if a sample is known or novel, a false familiarity (FN) attack is applied to novel samples to increase MLS values. A false novelty (FP) attack is applied to known samples to reduce MLS values.

The analysis found that in uninformed settings, as shown in Fig. 2(a) and 2(b), the FP attack is more effective in destroying the rankings than the FN attack. From Fig. 2(c) and 2(d), the paper explained this result with the hypothesis of Dietterich & Guyer. FN attack might effectively increase MLS of both known and novel samples which preserves the familiarity ranking of samples. On the other hand, an FP attack might be more effective in decreasing the MLS of known samples which destroys the familiarity ranking of samples. In informed settings, the research found that the FN attack is more effective in destroying the rankings (shown in Fig. 3). The paper concluded that the results confirm the hypothesis of Dietterich & Guyer.

**Strengths:**

Due to the real-world application of machine learning models, where it is common to encounter samples from classes unseen during training, OSR is a significant problem. This paper investigated adversarial attacks on OSR. This research was started by a good intuition motivated by the prediction of Dietterich & Guyer. The work tried to investigate several problems regarding adversarial attacks on OSR: relative effectiveness of False Familiarity and False Novelty attacks, one-step versus iterative attacks, benefits of informed attacks, and adversarial attack-based OSR score. Investigated losses for adversarial attacks seem reasonable as well as the proposed Adversarial reaction score (ARS) for OSR. The overall structure of the paper was good.

**Weaknesses:**

One of the major claims in the paper is that it confirmed the prediction of Dietterich & Guyer: it will be easier to familiarity score than to decrease it (as later requires changing logit/SoftMax of "all" known classes). In lines 268 to 286, the paper derived this from the uninformed attack situation. However, as uninformed attacks are applied on both novel and familiar samples, the majority of MLS change may occur in either novel or familiar samples (not both). While it is counterintuitive, it is still possible that when epsilon is small (around 0.2), most of the MLS increase in FN attacks happened only in familiar samples. In that case, the speculation in lines 271-274 makes less sense. (This speculation is likely correct, but the shown results do not "confirm" the hypothesis of Dietterich & Guyer.) Moreover, when epsilon was large (at least 0.8) in FN attack, Fig. 2(a) and 2(c) show a puzzling phenomenon where MLS was almost recovered while AUROC was low. This phenomenon seems to contradict the claim of ease in increasing the logit score. Because of these issues, we need MLS change plots for novel samples and the corresponding plots, separately, for familiar samples to properly conclude the claim. (Specificity and sensitivity plots for various epsilons can also be helpful. Less important figures like Fig. 2(e), 2(f), and Fig. 3 can be moved to the appendix instead.)

For the informed attack situation, the paper used only one epsilon value for each setting. Moreover, larger epsilon values were used for the FN attack than for the FP attack. Hence, it is hard to judge if the FN attack is more effective than the FP attack in informed attack settings. To compare the effectiveness of two attacks, it is better to show the results from at least 2 epsilon values with the same epsilon values for FP and FN attacks.

While the paper intuitively explained (Fig. 1 and lines 61-65) that False Novelty attack might be harder as it requires to reduce logits (or SoftMax prediction) for "all (known) classes" (unlike False Familiarity attack where changing "one class" is enough), there is no experiment on the effect of the number of known (familiar) classes.

While it is not necessary, identifying the relationship between the number of known (familiar) classes and the relative efficiency of FP/FN attacks in the informed setting can greatly strengthen the paper.

It can be confusing for the readers that MLS outperforms MSP in OSR performance (in lines 125 to 128) given their equivalence except for the magnitudes information in logits (as explained by Vaze et al.). It would be nicer if the paper briefly explained how MLS could outperform MSP.

It would be nice if the goal of the attacks (destroying ranks of samples) were explained when uninformed attacks were first introduced (lines 210-212).

While the paper used a non-traditional approach (RPROP) for iterative adversarial attacks, the paper did not provide or mention any plan for releasing their code. Attack hyperparameters, such as the number of steps, are also missing. I would recommend releasing the code for reproducibility.

*Minor comment (not affect rating)
While the title and abstract (lines 11-12) seem to indicate that the paper analyzed both familiarity-based Open-Set Recognition (OSR) approaches (MSP+MLS), the paper only experimented with one (MLS) of them.

In line 139, there are two periods (typo).

The terms "False Familiarity (FN)" and "False Novelty (FP)" attacks are confusing, especially in uninformed settings, as these are applied to both novel and familiar (known) samples. For instance, applying a "False Novelty (FP)" attack on "a novel sample" might result in "a perturbed novel sample" (it is a novel sample, but not a "False novel" sample). Perhaps, terms like "Familiarity (F)" or "Novelty (N)" attacks might be more appropriate in uninformed settings. Personally, using "F" and "N" for abbreviations is less confusing than using "N" and "P".

The real class $y$ is actually not used in OSR attacks, unlike closed-set attacks. Hence, y can be removed from OSR attack losses. For instance, $L_{max}(\theta,x,y)$ can be written as $L_{max}(\theta,x)$.

The x-axis of Fig. 2(a), 2(b), 2(c), and 2(d) seems to represent the "relative size of epsilon" rather than the "(actual) size of epsilon".

In lines 311 to 315, it seems the author(s) mistakenly swapped "FP" and "FN" attacks.

In equation (9), the shown formulation can use a different class for $f(x^{adv})$ and $f(x)$. Is it intended or a mistake (perhaps, intended for $f(x^{adv})\_{y^*}-f(x)\_{y^*}$ where $y^*$ is $\text{argmax}\_{y} f(x)$)? Which formulation was used in the implementation?

The changes in MLS are not monotonic in Fig. 2(c) and 2(d). Can the author(s) explain possible reasons for this? Perhaps due to linear attack (FGSM)?

**Final Rebuttal Confidence:**

4

**Final Rebuttal Justification:**

The adversarial attacks on OSR are problems that are worth investigating. Moreover, investigating the Familiarity Hypothesis is also important research regarding OSR. The research tried to answer several questions about adversarial attacks on OSR.

Edit: Previously, I had concerns regarding the correctness of the claims (hypothesis of Dietterich & Guyer: the (maximum) logit score can be more easily increased with an adversarial perturbation than decreasing it, fairness of comparing FP and FN efficiency in the informed setting). The author(s) addressed these concerns through updates and responses. I am now leaning toward the acceptance of the work.

**Justification:**

The adversarial attacks on OSR are problems that are worth investigating. Moreover, investigating the Familiarity Hypothesis is also important research regarding OSR. The research tried to answer several questions about adversarial attacks on OSR.
However, I have concerns regarding the correctness of the derivation of the main claims of the paper, including the confirmation of Dietterich & Guyer’s prediction that the (maximum) logit score can be more easily increased with an adversarial perturbation than decreasing it. These concerns might be easily resolved by providing detailed results (more separated investigation of known and novel samples). Hence, I am leaning toward the rejection of the paper at this stage, but this can be changed when issues are properly addressed later.

---

> ### Author Rebuttal · Authors · 2024-10-25
>
> Thank you very much for the constructive feedback which helped us to improve the paper.
> Below, we comment on the major points and hope that the additional analyses shown in the updated pdf (new Figure 3), helps to resolve any concerns.
>
> Note: The current pdf is not the final version yet, we will continue to incorporate the feedback for improvements for the camera ready version. Thanks!
>
> **Confirmation of the hypothesis of Dietterich & Guyer:** We respectfully disagree with the comment that our results do not confirm the prediction that it should be easier to increase the familiarity of one category rather than decreasing the scores of all categories.
>
> We reformulate this prediction into the question of: What type of attack is more effective? Our results show that the answer to which type of attack is more effective depends on the setting, which was not specified by Dietterich & Guyer. For the _informed attack_: Indeed, FN attacks are more effective (See old Fig. 3 / new Fig. 4), i.e. increasing the familiarity. In contrary, for the uninformed attacks, the FP attacks are more effective in reducing the AUROC (see Figure 2a,b), i.e decreasing the familiarity of all categories to destroy the ranking of OSR.
>
> To support our argument in line 271-274 (original submission), we visualize the median MLS separately for the familiar and novel samples in the new Fig. 3 in the new uploaded pdf. In new Fig 3a, we see that the FN attack increases the scores of both the familiar and novel samples similarly. This is not the case for the FP attack (see new Fig. 3b).
>
> **Large epsilon and non-monotonic MLS:** We hypothesize that with very large epsilon, the adversarial perturbations are turning the images into pure noise images. These noise images are far away from the original data distribution, leading to a covariate shift, which can cause very noisy predictions and OSR scores. This can result in a random ranking of the test samples as measured by the AUROC, while MLS behaving in a non-monotonic (random) behaviour. However, these large epsilons do not provide a meaningful subtle adversarial attack that could be undetected. We will extend our discussion to explain the issue with large epsilon.
>
> \*\*MISC\*\*
>
> **Informed attack setting:** We use the optimal epsilon for each attack informed by Figure 2 and also shown in new Figure 3 for the informed attack.
>
> **Goal of adversarial attacks:** Good point, we now state this explicitly at the beginning of Section 2.4.
>
> **Notation:** We use the same notation introduced for the FGSM attack by Goodfellow et al. (2015). Note that y corresponds to the target, i.e. the closed-set categories, and not the binary-label familiar vs. novel. Familiarity-based approaches do not rely on any I/OOD samples during the training and our adversarial attacks also do not rely on these binary labels.
>
> **Fig 2:** No, the epsilon is always given in absolute values.
>
> **lines 311 to 315:** Yes, thanks for spotting this mistake, where we mistakenly swapped "FP" and "FN" attacks.&#x20;
>
> **ARS computation:** In Equation 9, we intend to show that different classes can be used. We added a sentence to clarify that this can happen.

---

### Official Review · Reviewer_dRG9 · 2024-10-03
**First Review**

**Confidence:** 4

**Summary:**

The authors investigated the concept of adversarial Out-of-Distribution (OOD) examples [1]. More in detail, the paper focuses on two OOD detection methods (Maximum Softmax Probability (MSP) and Maximum Logit Score (MLS)) and gradient-based adversarial attacks (FGSM and BIM). The aim is generating adversarial In-Distribution examples (false negatives), wrongly accepted by an OOD detector, and adversarial OOD examples (false positives), with the reverse effect. The authors proposed different definitions of attack towards either ID or OOD samples, and also validated strategy is best when an attacker does not know if a sample is ID or OOD. Last, the adversarial OOD attack was suggested as a strategy to improve OOD detection, however the benchmark did not improve over the baseline (MLS).


[1] "Adversarial OOD examples are constructed w.r.t the OOD detector, which is different from the standard notion of adversarial examples (constructed w.r.t the classification model).", from Chen, Jiefeng, et al. "Atom: Robustifying out-of-distribution detection using outlier mining." Machine Learning and Knowledge Discovery in Databases. Research Track: European Conference, ECML PKDD 2021, Bilbao, Spain, September 13–17, 2021, Proceedings, Part III 21. Springer International Publishing, 2021.

**Strengths:**

The paper is clear, and the experimental setup is based on established OOD benchmarks [1], which enhances reproducibility. In my view, the paper's most significant finding is that when an attacker is unaware of the OOD detector's output, targeting false positives results in a more severe drop of performance compared to focusing on false negatives.

[1] S. Vaze, K. Han, A. Vedaldi, and A. Zisser- man. “Open-Set Recognition: A Good Closed- Set Classifier is All You Need”. In: Interna- tional Conference on Learning Representations (ICLR). 2022.

**Weaknesses:**

- The main weakness is that the concept of adversarial ID/OOD is not novel [1,2,5], therefore the authors should reference and address existing baselines. In [1,2] adversarial OOD/ID have been defined for both softmax and distance based OOD detectors, with stronger white- and black-box attacks, by also introducing approaches to make detection more reliable.
- The authors proposed a white-box attack, therefore it seems to me an unrealistic constraint to suppose that the attacker has access to the model weights but not the ID/OOD label (that the attacker can compute given the OOD output).
- The ARS does not improve over the baseline, even in a scenario where (correct me if I am wrong) ARS hyperparameters have been selected on the test set (instead of a validation set). In this regard, a previous similar approaches, considering the difference between normal and perturbed logits for detection, could help the authors improve their work [3].
- The experimental setup is quite limited (one model, one ID dataset).
- The literature review on generalized OOD detection should be updated with more recent benchmarks [4].

[1] Chen, Jiefeng, et al. "Atom: Robustifying out-of-distribution detection using outlier mining." Machine Learning and Knowledge Discovery in Databases. Research Track: European Conference, ECML PKDD 2021, Bilbao, Spain, September 13–17, 2021, Proceedings, Part III 21. Springer International Publishing, 2021.

[2] Chen, Jiefeng, et al. "Robust out-of-distribution detection for neural networks." arXiv preprint arXiv:2003.09711 (2020).

[3] Roth, Kevin, Yannic Kilcher, and Thomas Hofmann. "The odds are odd: A statistical test for detecting adversarial examples." International Conference on Machine Learning. PMLR, 2019.

[4] Zhang, Jingyang, et al. "Openood v1. 5: Enhanced benchmark for out-of-distribution detection." arXiv preprint arXiv:2306.09301 (2023).

[5] Azizmalayeri, Mohammad, et al. "Your out-of-distribution detection method is not robust!." Advances in Neural Information Processing Systems 35 (2022): 4887-4901.

**Final Rebuttal Confidence:**

4

**Final Rebuttal Justification:**

As I mentioned in my original review, I believe the paper has a clear presentation. Initially, I had concerns regarding existing frameworks for ID/OOD adversarial examples. However, I reconsidered this given that the authors specifically focused on Open Set Recognition (OSR), whereas prior work concentrated mostly on OOD. My second concern was on the significance of the results, that the authors have improved (or will improve in the camera-ready) by better explaining the experiments and by including additional analysis (Novel vs Familiar MLS, multiple $\epsilon$-values for the informed attack). My final opinion is inclined toward accepting the work.

**Justification:**

The main concern with the paper is that the authors did not address relevant literature on adversarial ID/OOD attacks. Additionally, the results remain preliminary, as they are restricted to softmax OOD detectors and gradient-based attacks, and fail to offer substantial contributions, such as improved robustness in OOD detection.

---

> ### Author Rebuttal · Authors · 2024-10-25
>
> Thank you very much for the constructive feedback which helped us to improve the paper.
> Below we answer the open questions and try to clarify a few points.
>
> **Related work:** Thank you for pointing us to these works. We will extend the discussion of related work on adversarial attacks to more general out-of-distribution (OOD) detection settings for the camera ready version. Our work focuses on familiarity-based open-set recognition (OSR), which is a special case of OOD. One crucial aspect, is that familiarity-based OSR approaches neither use OOD data during the training, nor introduce an explicit class/logit for the “openness” of the test data. Consequently, this affects the design of adversarial objectives. We will discuss these specific cases and contrast it with prior work on general OOD, which often uses various different definitions of distribution shifts.
>
> **White-box:** We would like to clarify that we study the adversarial vulnerability of OSR in both settings, uninformed and informed (i.e. having access to the binary label familiar vs. novel). However, this requires having access to the true label (ID/OOD) which is not the same as the predicted label.
>
> **ARS:** Thanks for pointing out this related work. While the evaluated adversarial reaction score (ARS) works on par with the MLS score, it is approaching OSR scoring from an adversarial point of view. We report these results because we believe that they contribute to a better understanding of the OSR problem, which might lead to the development of better scoring rules in the future.
>
> **Contributions:** We would like to clarify our contributions, which answer the formulated research questions (line 083-094):
>
> 1. We formulate two types of attacks (_false familiarity_: increase the max score over classes, and false novelty: decrease the scores of all classes) on open-set recognition, and show that their effectiveness depends on the setting.
>
> 2. We show that iterative attacks are more effective than the simple FGSM baseline, reducing the AUROC by an order of magnitude.
>
> 3. We show that informed attacks, i.e. assuming the knowledge of the binary label familiar vs. novel during the attack, can reverse the OSR ranking almost perfectly.
>
> 4. Finally, we show that the reaction to adversarial perturbation is different for familiar and novel samples, which can be used as an OSR scoring rule.
>
> Please note that this work does not propose an improved robustness for OSR. While the adversarial reaction score (ARS) works on par with the MLS score, it is approaching OSR scoring from an adversarial view. We believe that this observation might lead to better OSR approaches in the future.

---

### Official Review · Reviewer_EGqu · 2024-10-06
**Interesting experimental analysis into (a) best practices for and (b) comparisons between adversarial attacks for open set recognition.**

**Confidence:** 3

**Summary:**

This paper performs an analysis into adversarial attacks for open set recognition (OSR)†, that is methods that can trick ML classification systems into either believing an in-distribution class is actually out-of-distribution (called a false novelty or FP attack) or vice versa (called a false familiarity or FN attack). This paper first introduces these concepts (section 1) and attack methods (mainly variants of existing adversarial attack methods; section 2), before then performing experiments (section 3) assessing the following aspects of OSR attacks:
1. Whether it's easier to perform an FP attack or FN attack (Figures 2–3), which is perhaps linked to how the OSR approach actually works (i.e., whether it looks for the presence of "unusual features" or the absence of "usual features").
_results: FN attacks are easier (when the attacker knows whether the image belongs to closed or open set)._
2. How iterative attacks compare to single step attacks (e.g., FGSM [10]) in effectiveness (Figure 3).
_results: iterative attacks are found to be more effective and almost able to perfectly attack the classifier under consideration._
3. How uninformed attacks compare against informed attacks (an uninformed attacker does not know whether the input image is currently in or out of distribution, or in other words part of the closed set).
_results: an attacker can do much better in an informed attack (as expected), but perhaps more surprisingly this affects FN attacks more than FP attacks._
4. Can downstream effects of adversarial attacks be used as an OSR method (Section 2.6 and Figure 4).
_results: yes, but not sufficiently better than the max logit approach, which is a much simpler method._



† The OSR is done here by looking at the magnitudes of the logits of a classifier. However, the paper also investigates whether one can use downstream effects of adversarial attacks as an OSR method itself (Section 2.6 of paper and point 4 above).

**Strengths:**

I have broken down my view of the paper's strengths along the suggested axes of correctness, quality, clarity, and significance below.

### Correctness
The experimental setup and inferences made from the results seem reasonable, as well as the baseline model/weights chosen (based off of [3]). One small caveat is that only the TinyImageNet task is considered (rather than considering all of those in [3]).

### Quality
Overall the experiments seem interesting, although there are a couple of aspects which I think the paper could improve, which I have listed in the weakness section later.

Things that I think the paper did well:
* The paper's analysis may be helpful for understanding how OSR techniques work and aide their future development. To explain further: it is unclear currently how much OSR techniques make use of (a) "usual features" being missing versus (b) "unusual features" being present, when deciding whether an image is not part of the closed set. The relative ease of FN attacks in the informed setting suggests that (a) is the case, corroborating recent work [9].
* I also appreciated the fact that the authors properly evaluated the ARS approach for OSR, showing that actually it was not clearly more effective than existing approaches.


### Clarity
Overall, I thought the paper was clearly written and well presented. Figure 1 was very helpful in laying out the problem and the methods were well defined in Section 2. I only have some more minor feedback in terms of presentation:
* I think it would have been better to have devoted more space to informed attacks (as more important) and less to uninformed (see W1 below).
* I was not sure what the solid lines were in Figure 4b.
* I found the notation around losses/objectives (e.g., equations 5–7) a little confusing. Sometimes these are losses that are minimized and sometimes these are objectives that are maximized. This could have been made more consistent (e.g., by sticking to losses only and introducing negative signs as appropriate).
* Perhaps a little more explanation in Section 2.6 would have been helpful (I did not realize at first that this was introducing an approach for OSR rather than a way to measure attack effectiveness).

### Significance
I thought the paper contained several results interesting to  other members of the community:
* The paper's experiments (see quality section above) help explain how OSR attacks might work and best practices for currently carrying them out (which is needed to develop better defenses).
*  The strong success of iterative informed attacks (lines 298–321) may lead to more future research into preventing this.
* The fact that ARS did not work that well, means that although maybe not a method able to replace existing OSR techniques, it might spur more research into these ideas.

**Weaknesses:**

###  W1 More focus currently on uninformed attacks versus informed attacks
More focus seems to be on the uninformed attacks (figure 2) rather than the informed attacks (figure 3). This means that there is less relative space to analyze the informed attacks, which I think are the more interesting of the two. For instance, the difference between using different objective functions for the informed attacks is not shown in the paper's figures, which would have been nice to include.

### W2 Missing analysis into how attack's objective interacts with OSR method
Currently the different attacks are evaluated against only one OSR method (MLS). It is unclear whether the presented findings would apply to other methods (e.g., the mentioned MSP or even techniques using separate classifiers such as [7]), and so somewhat reduces the potential significance of the paper.



### Misc. Other Questions
Q1. Line 309 says "max loss" is better for FN attacks; however, from Figure 2a I would have thought Log-MSP is actually slightly better?
Q2: For FN attacks does the L2 Norm loss not have a similar limitation as to when it is used in the FP setting, which is explained on line 198 (i.e., that you can make a logit negative to increase this score without this affecting the MLS score and subsequent classification as not part of the closed set)?
Q3: I was a little confused that the MLS went back down in Figure 2c or back up in Figure 2d as $\epsilon$ increased, even though the AUROC continued to go down (i.e., the attack worked better from the attacker's perspective). Is there any intuition for this result? Is the same effect seen with the iterative attacks?
Q4: Figure 2c & d only show the median of the max logit score. What do the distributions of these scores look like? Presumably they are somewhat bimodal (due to the images from both the familiar and novel classes being included)?

### Minor typos (does not affect my score/review)
* line 104: "evaluate" -> "evaluated"
* Figure 2 captions on subplots: "advesarial" -> "adversarial"
* line 267: "uniformed" -> "uninformed"

**Final Rebuttal Confidence:**

3

**Final Rebuttal Justification:**

Having read the rebuttal and the other reviews, I still hold similar views to my original review. I thought the analysis into adversarial attacks for OSR was interesting, although limited in the OSR methods assessed (authors also acknowledge this). I thought the authors did a good job with the rebuttal and answered several of my questions, also promising to provide more information on the more interesting informed attacks. It was good to see one of the other reviewer's concerns seemingly resolved as a typo. I've gone with a lower confidence score due my familiarity with some of the related work.

**Justification:**

I think this paper provides an interesting analysis into adversarial attacks for OSR, comparing and finding best practices for choosing objectives and performing attacks. FN attacks are found to be easier in the informed setting, corroborating recent suggestions [9] that it may be easier to increase logit scores (and "usual features") than depress them. I think this information would be interesting for others and so have gone with a higher score, urging acceptance, although a lower confidence score due my familiarity with some of the related work.

---

> ### Author Rebuttal · Authors · 2024-10-25
>
> Thank you very much for the constructive feedback which helped us to improve the paper.
> Below we answer the open questions and comment on the mentioned weaknesses. Please also note the updated pdf with the new requested Figure 3.
>
> **W1:** With the comparison of uninformed and informed attacks, our goal was to show that this knowledge about the binary label (novel vs. familiar) is crucial when making predictions about the adversarial vulnerability of familiarity-based OSR methods. For example, Dietterich & Guyer made predictions about adversarial attacks without specifying what information is available to the attack. However, our analyses show that the conclusions depend on this discrimination.
>
> **W2:** We agree, that’s why we highlighted this limitation in line 357: “It remains to be tested if the observed adversarial robustness holds for alternative familiarity scores, such as the MSP.”
>
> **Q1:** Note that we look at Figure 2c (and 2d) showing the median MLS scores after the attacks to decide which is the best performing loss to evaluate the informed attacks. We do not use the AUROC plots (2a,b), because that metric evaluates the ordering after perturbing both the familiar and novel samples with the same attack. However, the max loss and Log-MSP perform roughly on par. We will rephrase that paragraph to clarify this.
>
> **Q2:** That is a good point. We will discuss the limitations of the L2-Norm on potentially negative logits in a dedicated limitations section. For the FN attack, where we aim to increase the max logit, it could indeed be an issue if the logit with the largest absolute value is negative. Empirically this seems not to be an issue in our experiments, but is indeed a limitation of the L2-norm that should be discussed.
>
> **Q3:** That’s a good point, which we will include in the discussion. With very large epsilon the adversarial perturbations are turning the images into pure noise images. These noise images lead to very noisy OSR scores that in turn lead to a random ranking of the test samples as measured by the AUROC. However, these large epsilons do not lead to a meaningful subtle adversarial attack that could be undetected.
>
> **Q4:** We have included the requested plots to show median MLS separately for the familiar and novel samples in the new uploaded pdf in Figure 3. Without adversarial perturbation, the distribution of the median MLS is indeed bimodal. As the familiar and novel samples react similarly to FN attacks (new Fig 3a) the distribution stays bimodal for a range of epsilons. However, for the FP attacks (new Fig 3b), the modes and distribution of familiar and novel samples converge to a similar distribution from around epsilon=0.2.

---

### Meta-Review · Area_Chair_UtMN · 2024-11-01

**Recommendation:** Accept (Oral)
**Confidence:** 4

**Metareview:**

## Paper Summary

The paper investigates the vulnerability of Open-Set Recognition (OSR) systems to adversarial attacks of two distinct types: 1) False Familiarity (or False Negative), when the objective is to lower the logit scores of familiar classes; and 2) False Novelty (or False Positive), which aims for an increasing on the logits of the novel class so that it is regarded as familiar to an OSR system. The authors also explore different ways of generating adversarial inputs (based on the Fast Gradient Sign Method and an iterative approach), as well as whether the attack is informed or uniformed, meaning that the OSR system may or may not know _a priori_ which type of attack it will deal with. Experiments conducted on the TinyImageNet dataset with diverse levels of adversarial perturbation showed that the logits can be easily increased with those adversarial perturbations, which leads to a higher effectiveness of FN attacks in the informed scenario. On the other hand, for the uninformed scenario, it was shown that FP attacks were more effective, as they destroy the original classification rankings by hiding familiar features, corroborating the Familiarity Hypothesis postulated by Dietterich & Guyer. Although there is no conclusion about which scenario (informed or uninformed) is better, the authors showed that iterative attacks are more effective than FGSM, which opens the door to further investigations on this specific modality. Finally, the authors proposed and dicussed an alternative metric for OSR systems called Adversarial Reaction Score (ARS), analyzing its potential correlations against well-established OSR score metrics, such as the Maximum Logit Score (MLS) and the Maximum Softmax Probability (MSP).

## General Comments

The paper looks well-written, with clear research questions and showcases a substantial variety of test scenarios, seeming to be pretty hard to make them fit in a 6-page manuscript. Even though the proposed ARS metric has not shown a significant improvement, some important questions have been raised about the possibilities of adversarial attacks and their respective effects under an open-set scenario, which could inspire other researchers in the field to explore new possibilities. Nevertheless, I believe the authors did a great job, especially after having addressed all the reviewers' suggestions. I hereby recommend the acceptance of this paper.

## Strengths

* Great paper organization, with very informative illustrations and examples, especially Figures 1 and 2;
* High variety of research questions, which were properly addressed along the manuscript;
* Highlights the potential perils of adversarial attacks based on image perturbations, which incentivates the research for more robust OSR methods.

## Weaknesses

* The concepts presented in the paper are not much novel;
* Absence of comparison against other networks, having only conducted tests on the VGG32 architecture;
* Adoption of ARS as a new OSR metric is still questionable, although there is room for improvements;

**Suggested Changes To The Recommendation:**

1: I agree that the recommendation could be moved down

---

### Decision · Program_Chairs · 2024-11-06

**Decision:**

Accept (Oral)

**Comment:**

We recommend an oral and a poster presentation given the AC and reviewers recommendations.